# Acetaldehyde in Polyethylene Terephthalate (PET) Bottled Water: Assessment and Mitigation of Health Risk for Consumers

**Andrea Re Depaolini, Elena Fattore** **, Giancarlo Bianchi, Roberto Fanelli and Enrico Davoli ***

Department of Environmental Health Science, Istituto di Ricerche Farmacologiche Mario Negri IRCCS, 20156 Milan, Italy; redepaolini.andrea88@gmail.com (A.R.D.); elena.fattore@marionegri.it (E.F.); giancarlo.bianchi@marionegri.it (G.B.); roberto.fanelli@marionegri.it (R.F.)
* Correspondence: enrico.davoli@marionegri.it; Tel.: +39-02-390141

**Abstract:** This study was designed to investigate the levels of acetaldehyde residues in polyethylene terephthalate bottled water and its significance in terms of consumer health. We analyzed 104 samples collected throughout Italy, so as to be representative of the national market. Parameters such as $CO_2$ level, shelf life, weight of the empty bottle and distance from the production sites to the point of sale were also collected. Although the levels of acetaldehyde complied with the limits established by Italian legislation, they varied widely, with concentrations ranging from 0.41 to 76.2 µg/L. An assessment of safety for human health, using the margin of exposure approach, showed that the amount of acetaldehyde in bottled water is unlikely to be of any safety concern for human health. The acetaldehyde residues were mainly due to $CO_2$ levels which influence solubility of acetaldehyde in water. They are also related to the size of the bottle and the distance from the store, but not to the shelf life, at least for 40 days. The findings suggest some good practices for a better product from the point of view of public health, like polymer quality and limitation of transport distances.

**Keywords:** water quality; acetaldehyde; PET bottles; public health; risk assessment; margin of exposure

## 1. Introduction

Acetaldehyde (AA) or ethanal is a volatile compound belonging to the large family of aldehydes, a class of organic chemicals containing a formyl group. It is one of the most important aldehydes, occurring widely in nature and produced on a large scale in industry. It is also naturally present in many non-alcoholic beverages and in foods (e.g., bread, coffee, ripe fruits, etc.) [1], as well as in the environment, since it originates from the metabolism of plants [2] besides being emitted directly into the atmosphere by combustion processes.

AA is a metabolite of ethanal, which is oxidized in the liver to acetaldehyde by alcohol dehydrogenase (ADH) then to acetate by aldehyde dehydrogenase (ALDH) [3]. In October 2009, the International Agency for Research on Cancer (IARC) and World Health Organization concluded that AA from an alcoholic beverage and formed from ethanal endogenously is a Group 1 carcinogen to humans [4], whereas the molecule by itself is classified as possibly carcinogenic to humans (Group 2B) because of inadequate evidence on humans. Cell cultures and animal experiments have shown that AA is a mutagen and carcinogen, since it can cause point mutations and form covalent bonds with DNA [5–9].

AA is also widely produced and used by industry, mainly as an intermediate in a large number of other chemicals and in the synthesis of flavor and fragrance acetals. It is also still voluntarily added in

food manufacturing (e.g., milk products, baked goods, fruit juices, candy, desserts, and soft drinks) and is the designated as a "generally recognized as safe" (GRAS) synthetic flavoring substance by U.S. Food and Drug Administration (FDA). AA at low levels gives a pleasant fruity aroma, but at high concentrations it has a pungent irritating odor [10].

AA is an unwanted food contaminant [11]. In alcoholic beverages, for example, it may be formed by yeast, acetic and lactic acid bacteria fermentation, or through auto-oxidation of ethanal and phenolic compounds [12]. Another source of AA contamination in food is polyethylene terephthalate (PET) which is widely used for food packaging and drink bottles [13]. PET can degrade under external factors such as sunlight and high temperature [14], and its main thermal degradation products are aldehydes, water, $CO_2$ and carboxyl end groups like carboxyl-terminated polyester chains but also terephthalic acid and monoglycol ester [15,16].

Carbonyl compounds residues in PET polymers have been described for almost 40 years [17–19] in water. It is one of the contaminants most commonly released from PET bottles during heating or during any type of thermally induced degradation [20] and its presence has been monitored in bottled water samples collected worldwide, in Japan, US and in European countries [13] and, recently, in several state-wide surveys [21,22]. All studies consistently report AA levels in PET bottled water, with higher levels of carbonated and higher storage time samples, demonstrating that this product is contaminated with residues migrating from the bottle and that contamination levels are regulated by multiple factors. In Italy, 14 billion liters of water were bottled in 2016 and the country has the largest European per-capita use of bottled water, with 206 L/year per capita; this is also the second largest in the world, after Mexico with 244 L/year [23].

Here we present the results of a nation-wide survey of AA residues in bottled water. The aim was to quantify AA levels in bottled water products sold on the Italian national market, to assess whether their concentrations pose any risk for consumers' health, and to investigate which factors mainly influence the migration of AA into water in order to define approaches that could mitigate the consumers' health risk.

## 2. Materials and Methods

We collected 104 PET-bottled commercial mineral water samples, either sparkling, still and natural effervescent in Italy from 24 producers. Samples were shipped to the laboratory directly by producers, on a voluntary basis, after request by Mineracqua, the association of manufacturers of bottled-water products. Sampling was representative of the whole country, including the major islands since we collected the main mineral water brands form each Italian region (Figure 1).

As soon as they arrived at the laboratory, the samples were placed in a cold room, stored at 4 °C, and analyzed within a few days. All samples were accompanied by a data sheet, compiled by the producer, giving information on the batch. The samples were organized into six groups, according to the sample type and size (0.5 or 1.5 L): natural still water 1.5 L (NW1.5), natural still water 0.5 L (NW0.5), natural effervescent 1.5 L (EW1.5), natural effervescent 0.5 L (EW0.5), sparkling carbonated 1.5 L (SW1.5) and sparkling carbonated 0.5 L (SW0.5).

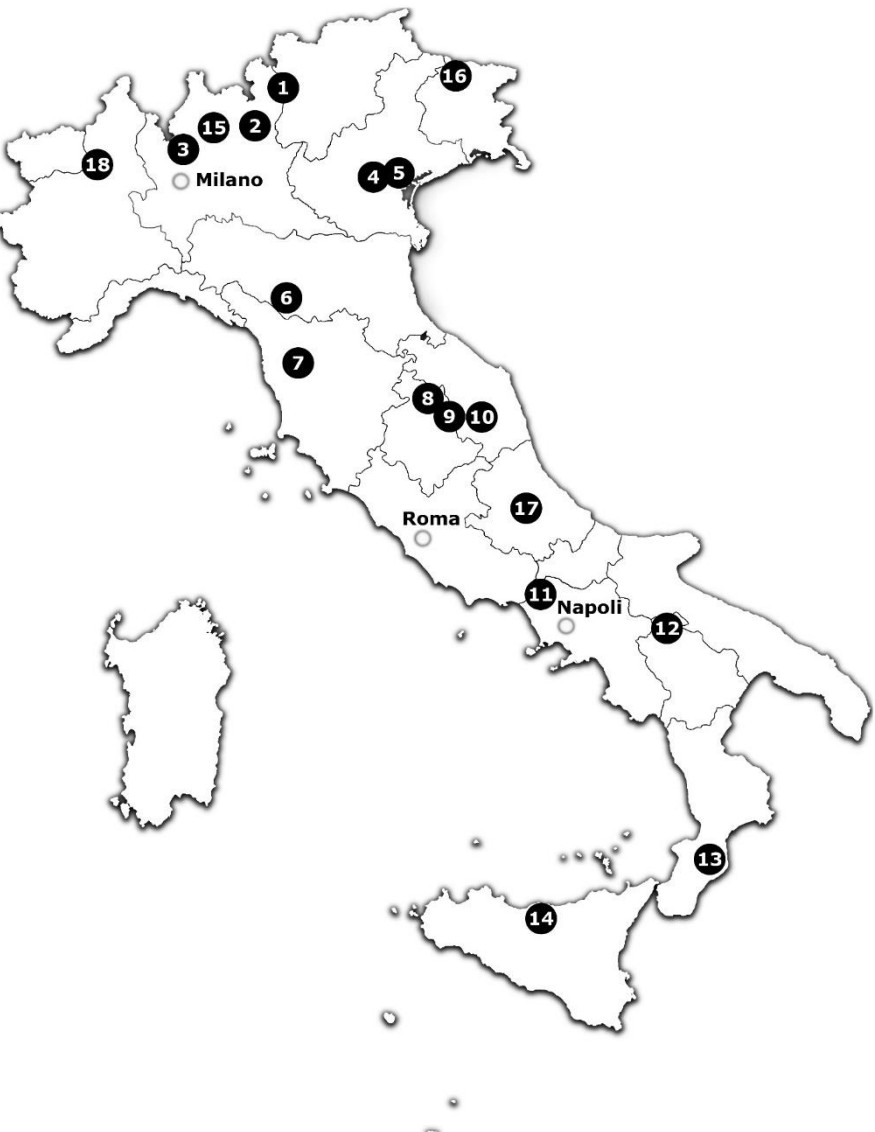

**Figure 1.** Map of Italy with the different sampling points. The black labels indicate the locations where samples were collected. The number inside the label indicates the number of different brand samples collected in each place.

The analytical procedure was a fully automated method, previously described [11]. It is based on head-space solid-phase micro-extraction (SPME) and on isotopic dilution, with acetaldehyde-d4 as internal standard. Briefly, 3 mL water were prepared in 10 mL headspace vials. After addition of 6 μL of a 50 ng/μL solution of internal standard solution, AA was derivatized directly in water samples into the thermally stable oxime product [24,25] by adding 20 μL of o-(2,3,4,5,6-pentafluorobenzyl) hydroxylamine hydrochloride (PFBHA) 10 μg/μL at 60 °C. The headspaces were extracted by solid-phase micro-extraction (SPME), with a 2-cm long tri-phasic fiber (Supelco, Bellefonte, PA) for 10 min. After extraction, the fiber was thermally desorbed in the GC injector, and the AA oxime derivative identified and quantified [26] against the isotopically labeled internal standard analog. Analysis was done by gas chromatography mass spectrometry (GC-MS) with an Agilent GC5890-MSD5975C system, using standard 70 eV electron ionization mode. The acquisition was Selected Ion Monitoring and full-SCAN mode (SIM/SCAN) in order to check for false positives due to contamination by unknown organic compounds. To quantify AA, ions with *m/z* 209 and 213, for native and deuterated internal standard respectively, were used, with a Varian CP-Select 624 CB chromatography column (60 m,

0.32 mm I.D., 1.80 μm film thickness). The GC oven temperature program was 80 °C for 1 min, 15 °C/min to 220 °C, held for 15 min. The initial pressure in the column was 29 kPa. Helium was used as a carrier gas at a constant flow rate of 0.8 mL/min. The injector operated in splitless mode and temperature was set at 250 °C. Transfer line temperature was maintained at 280 °C and ion source was set at 250°C. The linearity of the instrumental response was in the range of 1–600 μg/L with average regression coefficients of 0.9995. The limit of detection (LOD) and limit of quantitation (LOQ) was determined on the signal-to-noise ratio (S/N) of the lowest point in the calibration line. As an average LOD was 0.2 μg/L (with a S/N 3:1) and LOQ (with a S/N 10:1) was 0.7 μg/L. The coefficient of variation was 18 % (intra-day) and 13 % (inter-day).

Spearman's rank correlation analysis was used to investigate the correlation between AA concentration and the characteristics of the samples (size, $CO_2$ concentration, fixed residue, shelf life, time from sample collection to analysis, and distance from the production site). Kruskal–Wallis one-way analysis of variance (ANOVA) and Dunn's multiple comparison post-hoc test were used to assess differences in AA concentrations across the six mineral water groups. Statistical significance was assumed at probability <0.05. Projection to latent structures by means of partial least squares (PLS) analysis was used to assess the influence of the different variables on the AA concentration. PLS is the regression extension of the principal component analysis (PCA), which constructs new predictor variables, known as components, as linear combinations of the original predictor variables. The PLS analysis is suitable to model a response variable when the predictor variables are highly correlated or even collinear. Before the analysis, data were pre-treated by auto-centering and unit variance scaling. The software Graph Pad Prism Version 8.0.0 and Simca-P 11.0 package (Umetrics AB, Umea, Sweden) were used.

To simulate a possible effect of the agitation that samples receive during transportation on the AA concentrations in the samples, we have analysed a number of samples of natural and sparkling water, before and after controlled shaking for five days. The samples were placed in a shaker at 21 °C, with 35 agitations per minute. They were analysed before and after this treatment.

To estimate human exposure to AA, we used data provided by the former National Research Institute for Food and Nutrition (INRAN) [27] on bottled water consumption by the Italian population. The average daily dose of AA was calculated by multiplying the maximum concentrations of AA found in our samples by the corresponding daily consumption divided by the reference body weight. For risk assessment, we applied the European Food Safety Authority (EFSA)'s margin of exposure (MOE) approach recommended by the Scientific Committee [28], that considers safety concern arising from the presence in food of substances which are both genotoxic and carcinogenic [29]. The MOE is the ratio of the no-observed-adverse-effect level (NOAEL) or the lower 95% confidence limit of the benchmark dose (BMDL) (29,30), to the theoretical or estimated dose or concentration of human exposure. In this study, the MOE calculation considered a BMDL of 56 mg/kg BW per day, corresponding to a 10% increase in cancer incidence, as in experimental studies [30]. To simulate a worst-case scenario, we considered a daily consumption of bottled water equal to the 99th percentile for consumers only (1.85 L/day of bottled water), the highest concentration of AA found in this study and the conservative value for body weight of 60 kg.

## 3. Results

AA concentrations in the six groups of mineral water samples spanned more than two orders of magnitude (Figure 2), ranging from 0.41 to 76.2 μg/L. The highest concentrations (mean ± standard deviation) were detected in sparkling carbonated water of 0.5 L (35.1 ± 16.8 μg/L), followed by sparkling carbonated water of 1.5 L (27.8 ± 15.8 μg/L) and natural effervescent water of 1.5 L (6.05 ± 9.90 μg/L); the concentrations were lowest in natural effervescent water of 0.5 L, natural still water of 1.5 L and natural still water of 0.5 L, corresponding to 1.87 ± 1.79, 1.51 ± 1.77 μg/L and 0.915 ± 0.388 μg/L, respectively (Figure 2). ANOVA and the multiple comparison post-hoc test showed significant differences in the AA concentration between the sparkling water (0.5 and 1 L) and the other samples.

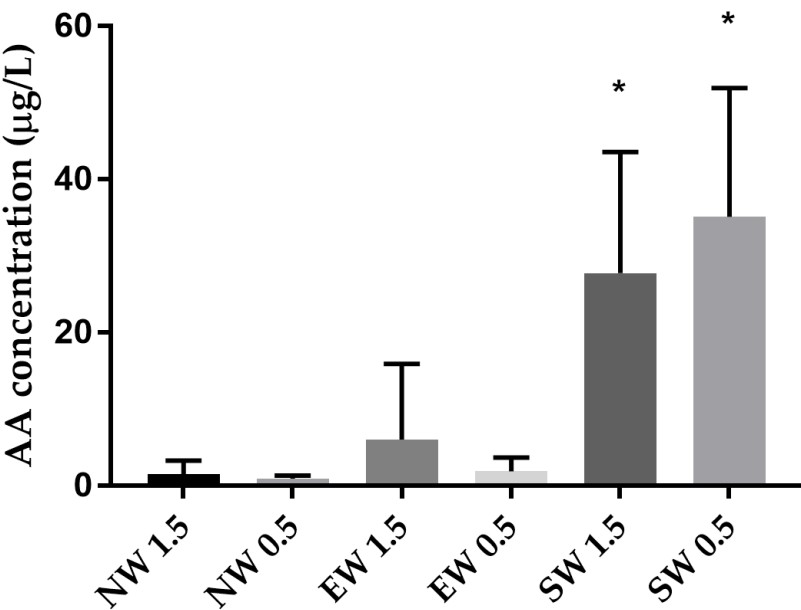

**Figure 2.** Acetaldehyde (AA) concentrations (mean ± standard deviation) for each group of water. (* Significantly different from the other groups (Kruskal–Wallis one-way analysis of variance and Dunn's multiple comparison post-hoc test, $P < 0.05$). NW1.5 = natural still water 1.5 L, n = 25; NW0.5 = natural still water 0.5 L, n = 17; EW1.5 = natural effervescent water 1.5 L, n = 4; EW 0.5 = natural effervescent water 0.5 L, n = 4; SW1.5 = sparkling carbonated water 1.5 L, n = 22; SW0.5 = sparkling carbonated water 0.5 L, n = 16).

Characteristics of the samples analyzed, grouped according to volume (0.5 or 1.5 L) and the $CO_2$ levels in sparkling, natural effervescent and still water are shown in Table 1. The correlation matrix between AA concentrations, volume and sample weight, $CO_2$ concentration, fixed residues, shelf life and the distance from the point of production showed that AA significantly positively correlated with the $CO_2$ concentration and the distance from the production site (Table 2).

**Table 1.** Characteristics expressed as mean and standard deviation (SD), of the samples analyzed.

| Sample [1] | No. | Weight (g) | | $CO_2$ (g/L) | | Fixed Residue (mg/L) | | Shelf Life (Days) | | Distance from Production Site (km) | |
|---|---|---|---|---|---|---|---|---|---|---|---|
| | | Mean | SD | Mean | SD | Mean | SD | Mean | SD | Mean | SD |
| NW1.5 | 25 | 26.3 | 3.4 | 0.078 | 0.390 | 192 | 148 | 37.1 | 63.6 | 91.5 | 132 |
| NW0.5 | 17 | 13.0 | 1.83 | 0 | 0 | 193 | 133 | 34.0 | 22.3 | 122 | 142 |
| EW1.5 | 4 | 34.1 | 3.13 | 2.11 | 0.643 | 1060 | 308 | 38.3 | 5.50 | 208 | 110 |
| EW0.5 | 4 | 17.7 | 2.87 | 2.11 | 0.643 | 1060 | 308 | 41.2 | 11.3 | 340 | 322 |
| SW1.5 | 22 | 29.4 | 2.96 | 5.86 | 0.895 | 182 | 124 | 30.7 | 20.6 | 87 | 110 |
| SW0.5 | 16 | 14.7 | 1.73 | 6.20 | 0.508 | 194 | 137 | 30.3 | 16.5 | 120 | 128 |

[1] NW1.5 = natural still water 1.5L; NW0.5 = natural still water 0.5 L; EW1.5 = natural effervescent water 1.5 L; EW0.5 = natural effervescent water 0.5 L; SW1.5 = carbonated sparkling water 1.5 L; SW0.5 = carbonated sparkling water 0.5 L.

**Table 2.** Correlations between acetaldehyde (AA) concentrations and other parameters of the mineral water samples.

| | Spearman Coefficient (*p*-Value) | | | | | | |
|---|---|---|---|---|---|---|---|
| | AA (mg/L) | Volume (L) | Weight (g) | $CO_2$ (g/L) | Fixed Residue (mg/L) | Shelf Life (Days) | Distance (km) |
| AA (mg/L) | 1.000 (0.000) | | | | | | |
| Volume (L) | −0.098 (0.327) | 1.000 (0.000) | | | | | |
| Weight (g) | 0.152 (0.135) | 0.831 (0.0001) | 1.000 (0.000) | | | | |
| $CO_2$ (g/L) | 0.741 (0.0001) | −0.034 (0.733) | 0.140 (0.169) | 1.000 (0.000) | | | |
| Fixed residue (mg/L) | 0.037 (0.711) | −0.028 (0.775) | 0.080 (0.432) | 0.059 (0.559) | 1.000 (0.000) | | |
| Shelf life (days) | 0.101 (0.318) | −0.106 (0.298) | −0.084 (0.409) | 0.006 (0.955) | 0.158 (0.117) | 1.000 (0.000) | |
| Distance (km) | 0.231 (0.022) | −0.145 (0.154) | −0.001 (0.996) | −0.031 (0.759) | 0.301 (0.003) | 0.554 (0.0001) | 1.000 (0.000) |

Figure 3 shows the results of the PLS analysis. The PLS score plot (Figure 3a) shows the samples separated mainly according to the $CO_2$ concentration along the first component. In particular, carbonated sparkling waters samples are positioned on the right side of the plot, the natural still waters samples on the left side, and the natural effervescent waters in an intermediate position. The PLS regression coefficients (Figure 3b), showing the weight of the predictor variables on AA concentration, show that $CO_2$, and, to a lesser extent, the distance from production to the point of sale, are the main variables directly influencing the AA concentration. The fixed residue, instead, was negatively related to the AA concentration, possibly because these waters generally have higher pH due to the bicarbonates.

Results from the 5 days agitation experiment, performed to simulate the effect of the shaking that occurs during transportation, showed that there was an average increase of AA in sparkling water samples of 17.0% compared to no increase in natural still water samples (Table 3). The AA concentration raised in all samples of sparkling water, although with non-constant increases. The natural still water samples showed no detectable changes in AA concentrations.

**Table 3.** Concentrations of acetaldehyde (AA) in samples subjected to shaking. A replicate of a 1.5 L sparkling water sample has been performed (A and B). [1] = not detected.

| | AA (µg/L) | AA Increase (µg/L) | AA Increase (%) |
|---|---|---|---|
| Natural still water 0.5 L not shaken | 0.5 | | |
| Natural still water 0.5 L shaken | 0.5 | nd [1] | nd |
| Sparkling water 0.5 L not shaken | 53 | | |
| Sparkling water 0.5 L shaken | 60.1 | 7.1 | 13.4 |
| Sparkling water A 1.5 L not shaken | 13.2 | | |
| Sparkling water A 1.5 L shaken | 14.6 | 1.4 | 10.6 |
| Sparkling water B 1.5 L not shaken | 9.3 | | |
| Sparkling water B 1.5 L shaken | 11.8 | 2.5 | 26.9 |

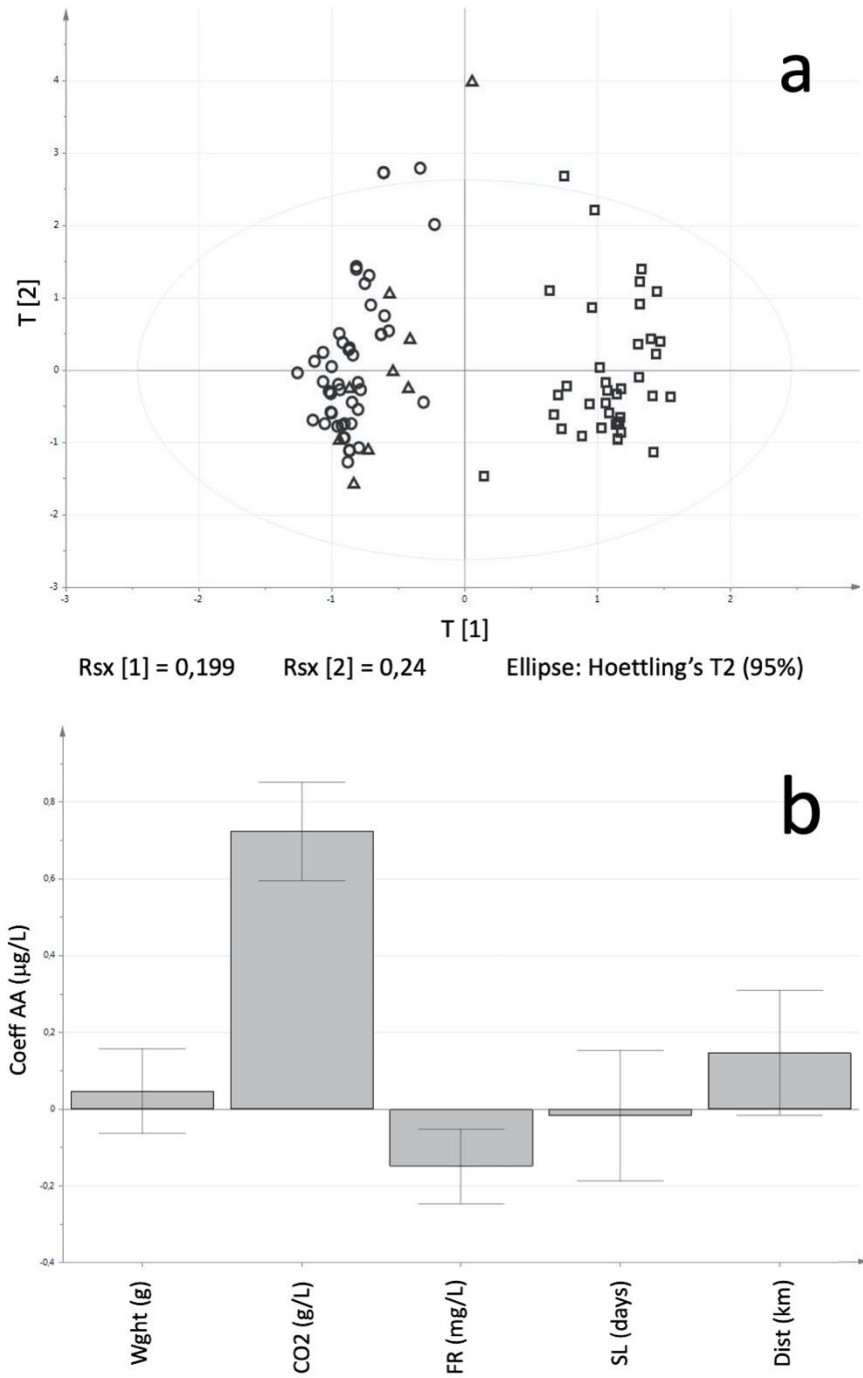

**Figure 3.** Results of projection to latent structures by means of partial least squares (PLS): (**a**) PLS score plot showing the samples in the space of the first (t[1]) and second (t[2]) principal component. Circles represent natural still waters, triangles the natural effervescent waters and squares the carbonated sparkling waters; (**b**) PLS regression coefficients, showing the influence of the different variables on the acetaldehyde concentration. (Legend: Wght = bottles weight, Dist = distance from production site, SL = shell life, FR = fixed residue).

Finally, the risk assessment for consumers' health resulted in a MOE low concern level of 24,000. This value was the result of the ratio between the BMDL of 56 mg/kg BW per day and the exposure dose of 2.3 µg AA/kg BW per day, derived from the daily consume of bottled water of 1.85 L/day, assuming an average BW of 60 kg, and the highest concentration of AA detected in this study of 76.2 µg/L.

## 4. Discussion

Different studies have reported AA concentration ranges similar to our analysis. Mutsuga et al. [13] found AA levels in PET bottles from Japan, Europe and North America ranging from 44.3–107.8 µg L$^{-1}$, from (35.9–46.9 µg L$^{-1}$ and from 41.4 to 44.8 µg L$^{-1}$, respectively. Although the range of AA concentrations found in this study is wide, they are below the Italian legal limits [31].

Results of correlation matrix seem to confirm that AA comes from the polymer in which the water is bottled and it is extracted from the polymer, dissolving in the water, probably due to multiple factors. The diffusion of AA is affected by temperature, storage time, and the carbonation, and the resulting lower pH of bottled drinking water [13,32–34]. However, Darowska et al. [35] suggested that the pressure exerted by $CO_2$ on the PET wall could promote diffusion of AA. Lorusso et al. [36] observed that the presence of $CO_2$ is important for the release of AA from PET to water: the levels of AA in carbonated water had increased in samples stored for six months at room temperature and at 40 °C. Using the same experimental conditions and the same type of bottles, the AA was not detected if distilled water was used instead. Results of the simulation tests showed an effect of the distance from the production site on the AA concentration. As reported by Nijssen et al. [18], Nawrocki et al. [32] the lower pH associated with $CO_2$ increased the amounts of AA in water. After long-term storage (8 to 9 months) this contaminant decreases as the bottles are permeable to carbonyl compounds [32].

Darowska et al. [34] instead conclude that $CO_2$ alone was not responsible for the higher levels of AA in bottled water but the pressure exerted by the gas on the PET wall favoured the diffusion from the polymer to the water. Bach et al. [33] studied the role of pH only, showing that PET in contact with deionized water at pH 4.5 and 6.5 does not change the migration of AA. Ewender et al. [32] studied the diffusion at different times of AA in carbonated and non-carbonated mineral water in 11 PET bottles. Various evidences also show how the carbonation of water directly influences the diffusion and stability of AA in mineral water [36,37] and [31]. Thus, it is possible that during transport different factors act on the water bottles, such as agitation and different storage temperatures, which could increase the extraction efficiency, driven by dissolved $CO_2$ in water, on the PET, with consequent release of AA.

The safety of AA exposure was assessed using the MOE approach. This gives an indication of the level of concern. The EFSA Scientific Committee considers a MOE of 10,000 or more, based on animal cancer bioassays, of low concern from a public health point of view and, as a consequence, of low priority for risk management action [37]. The present risk assessment showed a MOE higher than 10,000. This means that the amount of AA found in bottled water is unlikely to be of concern for human health [28].

## 5. Conclusions

We can conclude that the concentrations of AA in representative samples of PET bottled water on the Italian market pose negligible health risk for the population. Migration of this contaminant in water, as expected, depends on the $CO_2$ concentration, probably due to the lower pH but also to the pressure exerted by the gas on the PET, as well as the distance from the production site to the retailer: this, causing agitation of the bottles and exposing them to different temperatures during the transport, leads to an increase in the extraction of AA by $CO_2$ from the polymer to the water.

In view of the health concerns related to this chemical, based on the recent IARC evaluations, its presence in our diet and the large consumption of PET bottled water, in our opinion acetaldehyde residues in water should follow the ALARA principle (as low as reasonably achievable). Good practice for a better product, such as polymer quality control and limitation of the transport distance should be recommended by the category associations for this product.

**Author Contributions:** Conceptualization, E.D. and E.F.; methodology, E.F. and E.D.; formal analysis, G.B. and A.R.D.; resources, E.D.; data curation, E.F. and A.R.D.; writing—original draft preparation, A.R.D. and E.F.; writing—review and editing, A.R.D., E.F. and E.D.; supervision, E.D.; project administration, E.D.; funding acquisition, R.F.; All authors have read and agreed to the published version of the manuscript.

**Funding:** This research was funded by Mineracqua with an unrestricted grant to Istituto Mario Negri IRCCS.

**Conflicts of Interest:** The authors declare no conflict of interest. The funders had no role in the design of the study; in the collection, analyses, or interpretation of data; in the writing of the manuscript, or in the decision to publish the results.

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
