# Peer review of "Acetaldehyde in Polyethylene Terephthalate (PET) Bottled Water: Assessment and Mitigation of Health Risk for Consumers"

_applsci, doi:10.3390/app10124321_

Round 1

Reviewer 1 Report

I would appreciate if the introduction consisted of more information about similar analysis published.

In the experimental chapter you have to write the ionisation mode, eg EI 70 eV or if you used CI or something else. I want also to know the temperature of the ion source.

The LOD and LOQ should be described how you found and the LOQ is important to know.

Please use the same unit when you can, eg the LOD is reported as 0.2 ng/mL and in the results you are writing eg. 0.41 to 76.2 μg/L. Doing this makes the manuscript more difficult to read.

In figure 2 I expect to see the number of replicates.

The document was send to Urkund for check of plagiarism, and the result was 18% similarity, including references. this is not alarming, but please rephrase som sentences.I found some text similarities in the following sections e.g.:

80% similarity: CO2 promoted the release of AA from PET to water: after six months of storage, levels of AA in carbonated water had risen in samples kept at room temperature and at 40°C, whereas AA was not detected in distilled water under the same experimental conditions and in the same kind of bottles. 208

ref:
https://hal.archives-ouvertes.fr/hal-00683918/

64% similarity:

The influence of CO 2 content in bottled water was also studied by Darowska et al. [34] concluding that the CO 2 alone was not responsible for the higher levels of AA in bottled water and that the pressure exerted by the gas on the PET wall promoted the diffusion. Bach et al. [33] showed that PET in contact with de-ionised water at pH 4.5 and 6.5 did not enhance the migration of AA. Ewender et al. [32] studied the short and long-term diffusion of AA into carbonated and non-carbonated mineral water in 11 refillable and non-refillable PET bottles. Carbonation of the water directly influenced the diffusion and stability of AA in mineral water as shown by Lorusso et al. [36] ,Porretta and Minuti [37] and Nawrocki et al. [31].

ref:

https://hal.archives-ouvertes.fr/hal-00683918/

Minor comments:

L10: change to «corresponding author»

L22-23: rephrase this sentence. Do you really mean that CO2 extracts acetaldehyde into water?  Or does in influence upon the solubility?

L31: use subscript of the numbers of hydrogens at the methyl group.

L31: why informe about the cas no in the introduction, put it into chapter of the experimental.

L56: can you be more precise about what kind of compounds that might be formed that you define as carboxylic and groups?

L97: mass to charge should be written in cursive

L99: use space between the number and degree Celcius

L100: use space between the number and degree Celcius and the rest of the paper.

L100: rephrase this sentence. 29 KPa should be 29 kPa. You have to write if you have constant pressure or flow especially when you have a temperature program. 0.8 ml/min should be written as 0.8 mL/min.

L104: use space between the number and %, and in the rest of the manuscript

L103: please don’t use zero as the number of linear range, use the lowest concentration that was used in this test.

L114-117: different font size….

L202: This sentence is very oral; many authors agree…. Does that mean there are some authors that do not agree to this?

Author Response

I would appreciate if the introduction consisted of more information about similar analysis published.

Thank you for the comment, a sentence has been added, line 61

“All studies consistently report AA levels in PET bottled water, with higher levels of carbonated and higher storage time samples, demonstrating that this product is contaminated with residues migrating from the bottle and that contamination levels are regulated by multiple factors. “

Also, in Discussion paragraph, lines 233-237, similar studies are reviewed. But there are few similar studies, most of the studies on AA are related to the migration from the polymer and studies on market surveys are limited to limited number of samples, coming from the market, but, to our knowledge, none presented such a comprehensive study regarding territory product availability.

2.

In the experimental chapter you have to write the ionisation mode, eg EI 70 eV or if you used CI or something else. I want also to know the temperature of the ion source.

We added the two following sentences:

Line 100 “… using standard 70 eV electron ionization mode”

and

Line 108 “… and ion source was set at 250°C”.

3.

The LOD and LOQ should be described how you found and the LOQ is important to know.

Yes, it is correct. Line 109: The sentence has changed as follows:

“The limit of detection (LOD) and limit of quantitation (LOQ) was determined on the signal-to-noise ratio (S/N) of the lowest point in the calibration line. As an average LOD was 0.2 µg/L (with a S/N 3:1) and LOQ (with a S/N 10:1) was 0.7 µg/L.”

Please use the same unit when you can, eg the LOD is reported as 0.2 ng/mL and in the results you are writing eg. 0.41 to 76.2 μg/L. Doing this makes the manuscript more difficult to read.

 We changed the unit in µg/L

  1.  

In figure 2 I expect to see the number of replicates.

We added this information in the Figure 2 caption

5.

The document was send to Urkund for check of plagiarism, and the result was 18% similarity, including references. this is not alarming, but please rephrase som sentences.I found some text similarities in the following sections e.g.:

80% similarity: CO2 promoted the release of AA from PET to water: after six months of storage, levels of AA in carbonated water had risen in samples kept at room temperature and at 40°C, whereas AA was not detected in distilled water under the same experimental conditions and in the same kind of bottles.

Thank you, we rephrased the sentence as follows:

Line 122: “… the presence of CO2 is important for the release of AA from PET to water: the levels of AA in carbonated water had increased in samples stored for six months at room temperature and at 40 ° C. Using the same experimental conditions and the same type of bottles, the AA was not detected if distilled water was used instead

ref:
https://hal.archives-ouvertes.fr/hal-00683918/

64% similarity:

The influence of CO 2 content in bottled water was also studied by Darowska et al. [34] concluding that the CO 2 alone was not responsible for the higher levels of AA in bottled water and that the pressure exerted by the gas on the PET wall promoted the diffusion. Bach et al. [33] showed that PET in contact with de-ionised water at pH 4.5 and 6.5 did not enhance the migration of AA. Ewender et al. [32] studied the short and long-term diffusion of AA into carbonated and non-carbonated mineral water in 11 refillable and non-refillable PET bottles. Carbonation of the water directly influenced the diffusion and stability of AA in mineral water as shown by Lorusso et al. [36] ,Porretta and Minuti [37] and Nawrocki et al. [31].

Rephrased in

Line 231: “Darowska et al. [34] instead conclude that CO 2 alone was not responsible for the higher levels of AA in bottled water but the pressure exerted by the gas on the PET wall favoured the diffusion from the polymer to the water. Bach et al. [33] studied the role of pH only, showing that PET in contact with deionized water at pH 4.5 and 6.5 does not change the migration of AA. Ewender et al. [32] studied the diffusion at different times of AA in carbonated and non-carbonated mineral water in 11 PET bottles. Various evidences also show how the carbonation of water directly influences the diffusion and stability of AA in mineral water [36], [37] and [31].”

ref:

https://hal.archives-ouvertes.fr/hal-00683918/

Minor comments:

L10: change to «corresponding author»

Yes, done.

L22-23: rephrase this sentence. Do you really mean that CO2 extracts acetaldehyde into water?  Or does in influence upon the solubility?

Again, we agree with the point. We rephrased it as follows:

“The acetaldehyde residues were mainly due to CO2 levels which influence solubility of acetaldehyde in water”

L31: use subscript of the numbers of hydrogens at the methyl group.

Yes, done.

L31: why informe about the cas no in the introduction, put it into chapter of the experimental.

OK, we felt that this information was not substantial and we just removed it.

L56: can you be more precise about what kind of compounds that might be formed that you define as carboxylic and groups?

Yes, we added a Line 56:

“… and carboxyl end groups like carboxyl-terminated polyester chains but also terephthalic acid and monoglycol ester [15,16].”

L97: mass to charge should be written in cursive

Yes, done. It is now in line 102.

L99: use space between the number and degree Celcius

Yes, done.

L100: use space between the number and degree Celcius and the rest of the paper.

Yes, done.

L100: rephrase this sentence. 29 KPa should be 29 kPa. You have to write if you have constant pressure or flow especially when you have a temperature program. 0.8 ml/min should be written as 0.8 mL/min.

The sentences have modified

Line 105: “The initial pressure in the column was 29 kPa. Helium was used as a carrier gas at a constant flow rate of 0.8 mL/min.”

L104: use space between the number and %, and in the rest of the manuscript

OK

L103: please don’t use zero as the number of linear range, use the lowest concentration that was used in this test.

We changed it in Line 109:

“ …in the range of 1–600 µg/L …”

L114-117: different font size….

OK, corrected.

L202: This sentence is very oral; many authors agree…. Does that mean there are some authors that do not agree to this?

We agree. We were discussing the fact that there is general agreement on AA formation due to factors like temperature, storage time and carbonation of water. Few papers investigated effects of pressure. The sentence has been changed in

Line 219: “The diffusion of AA is affected by temperature… “

Reviewer 2 Report

The paper presents an analysis of acetaldehydes presence in bottled water in Italy. The paper in interesting. However it is not well-written and presented and can be improved.

Specifically, the results and discussion sections are too short and except of providing graphs and tables, do not adequately discuss the results. The statistical analysis results are presented in the form of tables and figures, needing a thorough explanation. In the discussion section, although there is some discussion on the findings, the authors should further discuss their results compared to other studies in Italy or other countries. 

Figures 3 and 4 cannot be read. The letters are too small.

Line 203: please follow the guidelines for authors for the citations.

Lines 70-73: please provide more data on the sampling process.

Author Response

The paper presents an analysis of acetaldehydes presence in bottled water in Italy. The paper in interesting. However it is not well-written and presented and can be improved.

1.

Specifically, the results and discussion sections are too short and except of providing graphs and tables, do not adequately discuss the results.

In the results we summarized the data in tables and graphs and we choose to use a Discussion paragraph to discuss results. The result paragraph is rather concise but all the results are actually presented and in most of the cases they are presented as graph or table and also mentioned in the text.

However, we expanded a little the part of the results regarding the calculation of the “margin of exposure” for the consumers as follows:

Line 207:  “Finally, the risk assessment for consumers’ health resulted in a MOE low concern level of 24,000. This value was the result of the ratio between the BMDL of 56 mg/kg BW per day and the exposure dose of 2.3 µg AA/kg BW per day, derived from the daily consume of bottled water of 1.85 L/day, assuming an average BW of 60 kg, and the highest concentration of AA detected in this study of 76,2 µg/L.”

The statistical analysis results are presented in the form of tables and figures, needing a thorough explanation.

Thank you for the comment. We added two sentences to better explain the statistics and the results obtained. We hope that this will be clear enough for the readers, now.

Line 118: “Projection to Latent Structures by means of partial least squares (PLS) analysis was used to assess the influence of the different variables on the AA concentration. PLS is the regression extension of the Principal Component Analysis (PCA), which constructs new predictor variables, known as components, as linear combinations of the original predictor variables. The PLS analysis is suitable to model a response variable when the predictor variables are highly correlated or even collinear. Before the analysis, data were pre-treated by auto-centering and unit variance scaling. The software Graph Pad Prism Version 8.0.0 and Simca – P 11.0 package (Umetrics AB, Umea, Sweden) were used.”

Line 177:  “separated mainly according to the CO2 concentration along the first component. In particular, carbonated sparkling waters samples are positioned on the right side of the plot, the natural still waters samples on the left side, and the natural effervescent waters in an intermediate position. The PLS regression coefficients (Figure 3b), showing the weight of the predictor variables on AA concentration, show that CO2, and, to a lesser extent, the distance from production to the point of sale, are the main variables directly influencing the AA concentration. The fixed residue, instead, was negatively related to the AA concentration, possibly because these waters generally have higher pH due to the bicarbonates.”

In the discussion section, although there is some discussion on the findings, the authors should further discuss their results compared to other studies in Italy or other countries. 

This point has been asked also by the other reviewer.

A specific sentence has been added in the Introduction paragraph, in line 61

“All studies consistently report AA levels in PET bottled water, with higher levels of carbonated and higher storage time samples, demonstrating that this product is contaminated with residues migrating from the bottle and that contamination levels are regulated by multiple factors. “

Also, in the Discussion paragraph, similar studies are reviewed. Actually, there are few similar studies, most of the studies on AA are related to the migration from the polymer and studies on market surveys are limited to few samples coming from the market.

2.

Figures 3 and 4 cannot be read. The letters are too small.

Yes. While changing the fonts we decided also, since they were related to the same model results, to use just one figure.

The legend has been changed accordingly as follows:

Figure 3. Results of projection to Latent Structures by means of partial least squares (PLS): a) PLS score plot showing the samples in the space of the first (t[1]) and second (t[2]) principal component. Circles represent natural still waters, triangles the natural effervescent waters and squares the carbonated sparkling waters; b) PLS regression coefficients, showing the influence of the different variables on the acetaldehyde concentration.

Legend: Wght = bottles weight, Dist = distance from production site, SL = shell life, FR = fixed residue.”

3.

Line 203: please follow the guidelines for authors for the citations.

Yes, we corrected the citations as follows:

Now, Line 220: “[13][32-34]”

4.

Lines 70-73: please provide more data on the sampling process.

Thank you for pointing it out, it is very important and we did not state it. A line has been added.

Line 75: “Samples were shipped to the laboratory directly by producers, on a voluntary basis, after request by Mineracqua, the association of manufacturers of bottled-water products.”

Reviewer 3 Report

The article deals with current problems of bottled water quality and its impact on the human health.
The authors used both appropriate analytical methods as well as statistical methods.
The article contains minor editing errors, e.g. in line 114-117 - the font is reduced,

The axis descriptions in Figures 3 and 4 are illegible, these figures should be improvemed

Author Response

The article deals with current problems of bottled water quality and its impact on the human health.
The authors used both appropriate analytical methods as well as statistical methods.

1.

The article contains minor editing errors, e.g. in line 114-117 - the font is reduced

Thank you, it has been corrected

2.

The axis descriptions in Figures 3 and 4 are illegible, these figures should be improved

Yes. While changing the fonts we decided also, since they were related to the same model results, to use just one figure.

The legend has been changed accordingly as follows:

Figure 3. Results of projection to Latent Structures by means of partial least squares (PLS): a) PLS score plot showing the samples in the space of the first (t[1]) and second (t[2]) principal component. Circles represent natural still waters, triangles the natural effervescent waters and squares the carbonated sparkling waters; b) PLS regression coefficients, showing the influence of the different variables on the acetaldehyde concentration.

Legend: Wght = bottles weight, Dist = distance from production site, SL = shell life, FR = fixed residue.”

Round 2

Reviewer 2 Report

The authors took into consideration the reviewers' comments and revised their manuscript accordingly.

The paper can be accepted at its present form.